## [Peer review file · Nature Communications]

REVIEWER COMMENTS

Reviewer #1 (Remarks to the Author):

First, as I strong supporter of a more open review process, would also like to reveal my identity during the review process as Prof. Gary Steele. I hope that by including my own name, it has reminded me to formulate my review in as objective a way as possible, and (hopefully) free of emotional response. I believe that this, combined with more open communication, will result in a more efficient and fair review process for scientific work.

In the manuscript, the authors present measurements on a silicon nitride membrane mechanical resonator capacitively coupled to a superconducting microwave cavity. The device is assembled, as in previous work with such platforms, by a “flip chip” technique. The membrane itself contains a strain-engineered phononic shield which results in a “soft clamping” of the mechanical mode, as demonstrated in previous work, and dubbed “ultracoherent” due to the very high quality factor. They observe optomechanical coupling between the microwave photons in the cavity and the motion of the mode, and use sideband cooling to cool the device, reaching, based on their calibrations, an thermal phonon occupation number less than one (the “ground state”, following terminology used in the field).

The methodology, including the device concept, the assembly, the cooling techniques, the data analysis techniques, are well established, and cooling to thermal occupations less than one, although difficult, has been achieved now (relatively) routinely. In this sense, there is not too much new in this result. On the other hand, these experiments alone are already challenging, and integrating new elements, such as the large membrane-based phononic shield, always present unexpected challenges which are easy to underestimate. Furthermore, the result, in particular the possibility of cooling such a high-Q mechanical resonator to below unity occupation, is certainly significant and opens new avenues for research.

For this reason, in particular, that this is possible and has been achieved, I am inclined to give positive advice for publication in Nature Communications.

That said, however, the article is very low on the level of detail provided, both in how the experiments were implemented, and in how the calibrations were confirmed: if the only real value is the novelty that something like this is possible in this platform, then this information is, to me, of crucial importance.

In the text below, I include some major comments that are grouped according by topic in which I feel that would be required for publication in Nature Communications. In addition, there are also comments

on minor things which should be addressed. Should these all be addressed to a sufficient degree of satisfaction, I would be fine with recommending this work for publication in Nature Communications.

Representation of the literature

=====

The introduction of the manuscript does not accurately reflect the state of the art, and over-emphasises the importance of “soft clamping”. In particular, the following statement is a misrepresentation of the literature:

“For state-of-the-art electromechanical systems with $Q \sim 10^5$ operated at milliKelvin temperatures, typical coherence times are in the range of 100 microseconds [14, 10].”

In <http://arxiv.org/abs/1510.07468>, for example, achieved Q-factors on the order of 10^8 . Similar results were obtained (and at least presented at the Gordon Research Conference) by the Regal group at low temperatures with square membranes and silicon-chip phononic shields (I remember $>2 \times 10^8$ at < 200 mK), although I cannot find if these results were ever published.

While I strongly appreciate the novelty and benefits of the engineered soft clamping design, the benefit in parameters is in practice an improvement of only one order of magnitude (10^8 to 10^9), not the 3 orders of magnitude that the formulation of the introduction would lead the reader to believe.

I am also not really clear on the justification of the word “ultracoherent”? Is 10^9 really that much more “ultracoherent” than 10^8 ? It seems that when the authors use the words “ultracoherent”, what they really mean is “in-membrane strain-engineered phononic-shield”. Note that one does not a-priori imply the other. The delineation they have made in the literature discussion based on the use of “ultracoherent” terminology as a label in the way they do is inappropriate in the manuscript, and also in the title itself, and should be removed.

Furthermore, the last paragraph of the introduction then implies, unless one reads carefully and understands the above delineation based on their definition of “ultracoherent”, that no group has metallised a silicon nitride membrane, nor has one ever cooled a silicon nitride membrane resonator to the ground state, while, in fact, neither is true (although not both in the same experiment: one with “ultra” coherence that was not cooled to the ground state, and one “regular” coherence that was...).

Before accepting this work in Nature Communications, or any journal, this important issue should be addressed.

Insufficient data / information to support claims

=====

With the possibility of including supplementary information, there is no excuse for not including more of the basic characterisation data of the device. The microwave scattering parameters of the resonance should be plotted, along with an indication of the fits of the curves. There is no reason not to include this in an open data submission as well.

What are the dimensions of LC resonator? Actually where is it? It is not clear to me where the input, output, transmission line, and actual resonator are. What are the simulated values of L, C and R of the circuit? Are the associated values of κ , κ_{ext} and κ_{int} in agreement with those extracted from the microwave reflection measurements? A full schematic (cad drawing or at least an accurate sketch) of the chip should be included in the SI indicating such details.

Did the authors confirm that the cavity response is still linear at the highest powers? What was the critical photon number that they observed?

The authors specifically mention the relevance of dephasing in ultracoherent resonators in the introduction. However, they present only ring-down results, and, of course, it is not the energy decay time that is relevant but the resonator linewidth (including dephasing). Data from, eg, the FFT of a ringdown measurement (see eg <http://arxiv.org/abs/1510.07468>), or a swept spectral response measurement, should be included to convince the reader that there is no excess dephasing.

The authors ultimately point to cavity frequency noise for the limited cooling they achieve. Surely, it would be very easy for the authors to measure the broadband cavity phase noise: why is this not included? If it is mechanical in nature, eg due to cryostat vibrations, it is highly unlikely that it is white. It is also not clear if this cavity noise is power dependent: it can occur that in the nonlinear response regime at high power, superconducting cavities can generate excess phase noise. Is the cavity driven close to this limit during the cooling measurements? Is the noise floor increase they observe broadband or limited only the cavity linewidth? Again, these claims should be supported with data.

What is the expected / implied distance in the capacitor gap between the cavity and the membrane? This must be possible to calculate from their data, and should be included.

What is the design frequency of the mechanical damper? How is it constructed? It is easy to include for example a photograph of it in the SI.

Calibration

=====

In the manuscript, the authors have used a thermal noise peak measurement to calibrate their vacuum coupling rate and the final occupation they achieve.

More information should be provided on how this calibration is performed. In particular, in the range of 0 to 1 K, it is likely that the frequency and internal quality factor of the superconducting cavity has changed, and this must be accounted for in a calibration of the mechanical sideband power since it will change the transduction. To support the data in figure 2A, data showing cavity frequency, cavity internal and external linewidths, mechanical frequency, and mechanical linewidth should be plotted, along with representative fits at a selection of temperatures. Surely the authors have this data: it should be included. Also, datasets showing the observed thermal noise peaks should be presented, and, again, ideally uploaded as an open dataset (there is no reason not to).

More specific details on the exact calibration of g_0 should also be included. Was the calibration performed with resonant cavity driving? What was the amplitude of the cavity drive? With such a high Q-factor, strong resonant driving can easily become self-oscillation if the cavity frequency shifts slightly to the red during measurements, giving an incorrect thermal calibration. Presumably the authors took measures to mitigate this: those measured should be described.

Furthermore, while a thermal calibration is nice, relying only on the assumption that the sample is at some point in thermal equilibrium with the calibrated sensor, and that the analysis has been done correctly, the system itself is also fundamentally overconstrained: for example if one knows the noise floor of the 4K HEMT amplifier at its input reference plane, and makes an assumption about the attenuation between the sample and the HEMT, then one also has a full calibration of the sample power based on the RT noise floor observed in the spectrum analyser. In such capacitor gap cavities, the microwave resonance frequency itself is also a very sensitive measure of the capacitor gap, and also of g_0 if one performs the appropriate electromagnetic simulations. Combining a knowledge of g_0 and knowledge of the power levels reference to the device also uniquely determines the thermal occupation from a trace at a single temperature.

The question then becomes: are these calibrations consistent with each other? The only real unknown in the second calibration is the attenuation between the sample and the HEMT. If the authors take their thermal calibration as correct, what do they obtain for the attenuation between the sample and the HEMT? Is the number reasonable? This type of cross check is an important part of ensuring that the absolute numbers are trustworthy (which is important as people assign so much importance to a difference between 0.8 and 1.2...)

This reference plane noise calibration analysis is also important in understanding better the noise squeezing they observe: for example, do they observe saturation of their HEMT amplifier with strong pump tones? How big is the sideband noise power? How does it compare to the expected generator sideband noise power? Without the absolute power numbers and attenuation estimates, it is hard to a priori rule these out.

Minor comments:

=====

In equation 1, it is quite important to emphasise that the temperature T here is the temperature of the intrinsic `_bath_` of the resonator, not the resonator's mode temperature. In particular, cooling the resonator mode to uK temperatures with sideband cooling will not increase its thermal decoherence rate.

* n and n_{cavity} *: Figure 3: Please be clear on what the shown PSDs are. Do they correspond to microwave sideband power spectral densities? Or are they mechanical displacement spectral densities? If they are mechanical displacement PSDs, then express them in $[\text{length unit}]/\sqrt{\text{Hz}}$.

"-45 dBm at the source": For the reader this is an arbitrary number. Given that the authors have a thermal calibration, and also know the cooperativity for example, it should be easy for them to translate all numbers to powers referenced at the input port of the sample. All powers should be referenced to a more relevant reference plane (ie. input of sample).

"It has been corrected and is presented in Fig. 2A to show the thermalization of the mechanical oscillator to the base plate of the cryostat." The users should be more specific on exactly what correction has been applied to the data. Again, the raw data should be plotted, and there is no reason not to include it as open data.

I appreciate that the authors have uploaded a zipfile of the data plotted, even before publication. It would be nice if there was some more documentation of what that data was (a readme.txt for example). A very nice option would be to include code (jupyter notebooks for example) that demonstrate plotting the data. But this is already a nice step (although I did not myself have time to look at the data...).

Reviewer #2 (Remarks to the Author):

The manuscript presents an experimental report on the realization of ground-state cooling of an ultracoherent electromechanical system. Specifically, a microwave resonator is coupled to a soft-clamped silicon nitride membrane with a super-conducting metal pad. In addition to the ground state cooling, the major achievement here is to reach the strong quantum cooperativity regime and to be able to maintain a quantum coherence time longer than 100 ms, about three orders of magnitude more than known electromechanical systems. It is argued that the results can be of practical significance for quantum memory applications or ultra-high precision force measurements. More fundamentally, the device with such long coherence times is suggested for testing quantum collapse theories. Overall, the paper is well-written. The diagrams and the figures are clear, and their discussions are intuitively explained.

The paper reports a significant technical advancement for modern quantum technologies and fundamental tests of the foundations of quantum physics. It can be recommended for publication after the authors address the following comments.

1. A short discussion of quantum optomechanical Hamiltonian and how its parameters and field operators are related to the current physical system can illuminate a broader range of readers; otherwise, the paper looks too technical and more accessible to experts in its current form. This discussion and model Hamiltonian should be used to justify the single-mode approximation of the model or to discuss if the setup allows for multimode extension (and if so, would it be possible to cool two or mode vibrational modes for more interesting quantum low-lying states?)
2. The authors should make a more convincing discussion of what is practically and fundamentally more challenging in their ultrahigh coherent system relative to that of Ref. [6] to clarify the novelty and impact of their contribution. At the same time, the authors should emphasize the distinction of their main idea that allows for the strong coupling and cooling in the ultrahigh coherent electromechanical systems, which was not possible with earlier implementations of such systems.
3. The method in the experiment uses coherent fields in getting the ground state; can the authors characterize the quantum state in more detail? Only the mean number of vibrational phonons is measured in the experiment; how about the higher-order moments and more complete characterization

of the phonon distribution and coherence properties? Perhaps some theoretical estimations and justifications can be given instead of measurements for that purpose?

4. Can the authors clarify if their system has any potential for non-local nonlinear terms in its model Hamiltonian to make it more directly relevant to quantum collapse models? This sounds a bit ambitious fundamental application of their results but indeed interesting if possible. A review article may be cited on this topic in addition to the Ref. [36], too.

Ozgur E. Mustecaplioglu

Reviewer #3 (Remarks to the Author):

In the manuscript entitled “Ground State Cooling of an Ultracoherent Electromechanical System,” the authors describe a new membrane-based electro-optomechanical system that permits efficient coupling to high quality factor phononic crystal membrane modes using circuit-based cavity optomechanical techniques. Since the circuit that they use to parametrically couple to the defect modes has a linewidth (240 kHz) that is significantly narrower than their mechanical resonance (1.5MHz) they are able to operate this system in the side-band resolved regime. The authors then use this system to demonstrate optomechanical cooling of their long-lived phonon modes, driving the system very near its mechanical ground state ($n < 1$). This manuscript describes a new system that represents a significant advance for the field of electro optomechanics, as it permits quantum coherent access to long-lived phonon modes within membrane type systems. The experimental studies appear to have been conducted with a great deal of thought and care. I recommend publication, provided that the authors address the following comments and make some clarifications within the manuscript discussed below.

Comments:

In section 2.2, the authors indicate that above 200 mK the is insignificant, and the system is assumed to be thermalized with the cryostat, whereas below 200 mK back action becomes significant. It would be helpful to the reader to provide further background regarding the various parameters that are likely changing in the system. For example, is the cooperativity changing as a function of temperature? Does the mechanical quality factor change as a function of temperature? I assume that the thermal

conductivity continues to plummet for temperatures below 200 mK. It might be useful to include some discussion of this point in either method section or supplementary information.

Also, do the authors have other anecdotal evidence to support the notion that the membrane remains thermalized with the cryostat at temperatures around 200 mK? For example, does the superconducting transition temperature of the membrane suspended superconductor coincide with that of the superconductor that is normally anchored to the cryostat (without a membrane)? This additional information could also be useful to include within a supplementary information.

Figure 1 and the description surrounding the electro optomechanical device do not quite provide the reader with enough information. In particular, Figure 1 doesn't provide enough detail to understand the circuit geometry and the coupling method. I only understood the coupling method after reading the description in the main text a couple of times and then stumbling across the inset the bottom of figure 4 in the methods section. I think that it would be prudent to include a conceptual sketch of the circuit diagram indicating how the circuit couples to the defect mode in figure 1 along with the micrographs of the fabricated system. Otherwise, it is unclear how coupling is being performed (which is pretty darn important).

Question:

It is interesting to see that tunneling-state two level systems are limiting the quality factor of the resonator at low temperatures. I'm curious to know if the authors observe the same tunneling-state two level system signatures with and without superconducting metal within the defect mode. Any observations along these lines might also be useful to the scientific community.

Aside from these areas of improvement described in my comments above, the manuscript does an excellent job of describing key features of their experiments and communicating them to a broad audience.

I'm happy to support publication once the authors make some clarifications and additions described above.

REVIEWER COMMENTS

Reviewer #1 (Remarks to the Author):

First, as I strong supporter of a more open review process, would also like to reveal my identity during the review process as Prof. Gary Steele. I hope that by including my own name, it has reminded me to formulate my review in as objective a way as possible, and (hopefully) free of emotional response. I believe that this, combined with more open communication, will result in a more efficient and fair review process for scientific work.

In the manuscript, the authors present measurements on a silicon nitride membrane mechanical resonator capacitively coupled to a superconducting microwave cavity. The device is assembled, as in previous work with such platforms, by a “flip chip” technique. The membrane itself contains a strain-engineered phononic shield which results in a “soft clamping” of the mechanical mode, as demonstrated in previous work, and dubbed “ultracoherent” due to the very high quality factor. They observe optomechanical coupling between the microwave photons in the cavity and the motion of the mode, and use sideband cooling to cool the device, reaching, based on their calibrations, an thermal phonon occupation number less that one (the “ground state”, following terminology used in the field).

The methodology, including the device concept, the assembly, the cooling techniques, the data analysis techniques, are well established, and cooling to thermal occupations less than one, although difficult, has been achieved now (relatively) routinely. In this sense, there is not too much new in this result. On the other hand, these experiments alone are already challenging, and integrating new elements, such as the large membrane-based phononic shield, always present unexpected challenges which are easy to underestimate. Furthermore, the result, in particular the possibility of cooling such a high-Q mechanical resonator to below unity occupation, is certainly significant and opens new avenues for research.

For this reason, in particular, that this is possible and has been achieved, I am inclined to give positive advice for publication in Nature Communications.

That said, however, the article is very low on the level of detail provided, both in how the experiments were implemented, and in how the calibrations were confirmed: if the only real value is the novelty that something like this is possible in this platform, then this information is, to me, of crucial importance.

In the text below, I include some major comments that are grouped according by topic in which I feel that would be required for publication in Nature Communications. In addition, there are also comments on minor things which should be addressed. Should these all be addressed to a sufficient degree of satisfaction, I would be fine with recommending this work for publication in Nature Communications.

Representation of the literature

=====

The introduction of the manuscript does not accurately reflect the state of the art, and over-emphasises the importance of “soft clamping”. In particular, the following statement is a misrepresentation of the literature:

“For state-of-the-art electromechanical systems with $Q \sim 10^5$ operated at milliKelvin temperatures, typical coherence times are in the range of 100 microseconds [14, 10].”

In <http://arxiv.org/abs/1510.07468>, for example, achieved Q-factors on the order of 10^8 . Similar results were obtained (and at least presented at the Gordon Research Conference) by the Regal group at low temperatures with square membranes and silicon-chip phononic shields (I remember $>2 \times 10^8$ at < 200 mK), although I cannot find if these results were ever published.

While I strongly appreciate the novelty and benefits of the engineered soft clamping design, the benefit in parameters is in practice an improvement of only one order of magnitude (10^8 to 10^9), not the 3 orders of magnitude that the formulation of the introduction would lead the reader to believe.

I am also not really clear on the justification of the word “ultracoherent”? Is 10^9 really that much more “ultracoherent” than 10^8 ? It seems that when the authors use the words “ultracoherent”, what they really mean is “in-membrane strain-engineered phononic-shield”. Note that one does not a-priori imply the other. The delineation they have made in the literature discussion based on the use of “ultracoherent” terminology as a label in the way they do is inappropriate in the manuscript, and also in the title itself, and should be removed.

Furthermore, the last paragraph of the introduction then implies, unless one reads carefully and understands the above delineation based on their definition of “ultracoherent”, that no group has metallised a silicon nitride membrane, nor has one ever cooled a silicon nitride membrane resonator to the ground state, while, in fact, neither is true (although not both in the same experiment: one with “ultra” coherence that was not cooled to the ground state, and one “regular” coherence that was...).

Before accepting this work in Nature Communications, or any journal, this important issue should be addressed.

We thank the referee for raising this issue. Our goal in this introduction is to give an overview to the broader landscape of electromechanical systems. Whereas the body of work is too large for a comprehensive review, we agree with the referee that a more detailed description of the state-of-the-art may be helpful for a broad audience.

Regarding typical coherence times of state-of-the-art electromechanical systems, we have previously only referred to aluminum vacuum gap and quantum acoustic devices (Refs 10 and 14, respectively). We have now also added state-of-the-art results obtained with metallized silicon nitride devices (Refs 6 and 11, and new references Higginbotham et al, Nat Phys 2018, Delaney arXiv 2021 and Zhou et al, Nat Phys 2013) as well as piezoelectrically coupled nanophononic crystals (Refs 15, 17). Note, however, that in all these systems $Q < 10^7$ and coherence times are at (or below) the 1 millisecond range.

The reviewer rightfully points to one notable exception reported in 2015, in which a metallized silicon nitride membrane was found to have a very high Q factor of $\sim 10^8$. It should be noted, however, that this was obtained at particularly low temperature (~ 10 mK) and frequency (~ 100 kHz). Here, the scaling of $Q \sim 1/f$ (or even $1/f^2$ for soft-clamped devices) permits a high Q at the expense of low frequency, which makes the typically desired simultaneous operation in the overcoupled and resolved-sideband regimes more challenging. A very recent preprint obtained the same Qf product with a higher frequency [Delaney arXiv:2110.09539], i.e. a Q of $\sim 10^7$. To our knowledge, there is only one unpublished preprint that shows similarly high Q's (arxiv:1611.00878) at MHz frequency, but these were obtained with non-metallized membranes not directly applicable to electromechanics.

Nonetheless, we agree the 2015 result from the Steele group is sufficiently important to mention it explicitly.

The new paragraphs on coherence times of state-of-the-art electromechanical systems therefore now read as:

For state-of-the-art electromechanical systems operated at milliKelvin temperatures, typical Q-factors are $< \sim 10^7$ and coherence times are at most 1 millisecond. This applies to a wide variety of systems, including aluminum vacuum gap capacitors [18, 7, 19, 8, 9, 10], metallized silicon nitride membranes [6, 11, 20, 21] and strings [22], quantum acoustic devices [14, 16], as well as piezoelectrically coupled nanophononic crystals [15, 17]. As a notable exception, $Q \approx 10^8$ has been reported for a metallized silicon nitride membrane in 2015 [23]. Its enhanced performance over similar devices [6, 11, 20, 21] might be linked to its particularly low operation temperature (~ 10 mK) and frequency (~ 100 kHz)—which, among other things, can make operation in the simultaneously overcoupled and resolved-sideband regime challenging.

On the other hand, recent progress in the design of mechanical systems has allowed reaching quality factors in excess of 10^9 at Mega- to Gigahertz frequencies [24, 25, 26, 27, 28, 29]. At milliKelvin temperatures, such ultracoherent mechanical devices can reach $t_{\text{coh}} > 100$ ms, some two orders of magnitude beyond the typical performance of state-of-the-art devices (provided excess dephasing [27] is not an issue). However, so far, the mechanics' coupling to microwave modes has either been extremely weak [24], or absent because of lacking functionalization through e.g. metallization [25, 26, 27, 28, 29]. For this reason, these mechanical systems could not yet be harnessed in electromechanics.

Regarding the use of the term “ultracoherent”, we would like to clarify that there is no sharp delineation in the Q-factor, for when a system becomes ultracoherent. It is rather meant to indicate that these systems are at the current frontier of high coherence. (Similar to the widely-used term “ultrafast” laser, which invokes a clear but not a quantitative idea of its characteristically short pulse duration). At the time of publication, the above-mentioned work [23] could certainly also be called “ultracoherent”.

Furthermore, we observe that the term “ultracoherent” has been adopted by many groups in this sense in the recent literature (Ghadimi ... Kippenberg, Science 2018; Ivanov ... Deleglise, APL 2020; Galinsky ... Polzik, Optica 2020; Kosata ... Eichler, Phys Rev Applied 2020) including a recent paper in Nature Communications (Høj ... Andersen, “Ultra-coherent nanomechanical resonators based on inverse design”, 2021). In general, we therefore prefer to retain this term.

However, we have changed the last paragraph in order to avoid the possible misunderstandings pointed out by the referee. In particular, we mention explicitly that we follow earlier work in metallizing the membrane. For reasons of the text flow we do not repeat that ground-state cooling has been achieved before, this is already mentioned in the first paragraph. We also follow the referee in emphasizing once more that our ultracoherent mechanical system is, specifically, a soft-clamped silicon nitride membrane. The new paragraph of the introduction reads:

Here, we realize an ultracoherent electromechanical system based on a soft-clamped silicon nitride membrane [24]. Following earlier work [11, 5, 22, 6], we functionalize it with a superconducting metal pad. [...] In our work, we realize sufficient coupling strength to cool the mechanical mode to its quantum mechanical ground state. This implies that we have achieved a quantum cooperativity $C_q > 1$ [3] and heralds the possibility to deploy soft-clamped mechanical resonators for applications in quantum electromechanics.

Insufficient data / information to support claims

=====

With the possibility of including supplementary information, there is no excuse for not including more of the basic characterisation data of the device. The microwave scattering parameters of the resonance should be plotted, along with an indication of the fits of the curves. There is no reason not to include this in an open data submission as well.

What are the dimensions of LC resonator? Actually where is it? It is not clear to me where the input, output, transmission line, and actual resonator are. What are the simulated values of L, C and R of the circuit? Are the associated values of κ , κ_{ext} and κ_{int} in agreement with those extracted from the microwave reflection measurements? A full schematic (cad drawing or at least an accurate sketch) of the chip should be included in the SI indicating such details.

To clarify the geometry of the device, we have included a diagram of the loop-gap resonator as it is drawn in the lithography mask file, with dimensions (supplementary figure S1). A picture of the incoupling loop, which terminates the coaxial waveguide, used to inductively couple to the resonator is also included, as is a photograph of the flip-chip-device and the sample holder box.

We have also modified Figure 1 in the main text to improve clarity regarding the electric equivalent circuit and the device setup.

Finally, we have added finite-element simulation data in a figure S2 in the SI. We give the simulated geometry: a conducting loop on top of a silicon substrate held in place in the middle of a hollow volume; and the spatial profile of the microwave mode of interest. By varying the distance of the metallised pad (placed on the membrane) to the loop electrodes and fitting the corresponding microwave frequency, we can indeed extract other parameters of the resonator, such as the inductance and self-capacitance of the loop.

Did the authors confirm that the cavity response is still linear at the highest powers? What was the critical photon number that they observed?

In Fig. S8 we plot the cavity parameters as function of input microwave power (corresponding to the ground state cooling dataset). Since we do not see a strong dependence of the resonance frequency on microwave power, we exclude a significant cavity nonlinearity.

The authors specifically mention the relevance of dephasing in ultracoherent resonators in the introduction. However, they present only ring-down results, and, of course, it is not the energy decay time that is relevant but the resonator linewidth (including dephasing). Data from, eg, the FFT of a ringdown measurement (see eg <http://arxiv.org/abs/1510.07468>), or a swept spectral response measurement, should be included to convince the reader that there is no excess dephasing.

We thank the referee for pointing out this important aspect. We could not obtain high enough signal-to-noise directly in a FFT of the ringdown traces as suggested. However, it was possible to extract a (smoothed) time-trace of the instantaneous frequency during the ringdown, which showed only a very small linear drift on the order of 1.6 mHz over 600 s (figure S5A).

Additional strong evidence for insignificant excess dephasing comes from measurements with a second sample which we have now included in the supplementary information (figure S5B,C). These measurements display very good agreement between a linewidth and a ringdown measurement.

The authors ultimately point to cavity frequency noise for the limited cooling they achieve. Surely, it would be very easy for the authors to measure the broadband cavity phase noise: why is this not included? If it is mechanical in nature, eg due to cryostat vibrations, it is highly unlikely that it is white. It is also not clear if this cavity noise is power dependent: it can occur that in the nonlinear

response regime at high power, superconducting cavities can generate excess phase noise. Is the cavity driven close to this limit during the cooling measurements? Is the noise floor increase they observe broadband or limited only the cavity linewidth? Again, these claims should be supported with data.

In Fig. S4 of the SI, we plot a wide power spectrum at the mechanical sideband, for a high microwave power. The visible bump in the bandgap noise agrees with the frequency and linewidth of the cavity (measured independently by network analyser reflection). Thus the rising background noise seen in the mechanical spectra of Fig. 4 of the main text are filtered by the cavity. Following the analysis done in the supplementary of <https://arxiv.org/abs/1507.08898>, we may interpret this cavity noise as the relative phase noise between microwave pump and cavity.

We can also extract the value of this phase noise to be approx. -145dBc/Hz , which is about 10dB higher than the phase noise as specified (at 1MHz sideband frequency) by the manufacturer of our low noise signal generator. This discrepancy makes us suspect that the dominant phase noise actually stems from the cavity, such as cavity frequency jitter. Indeed we have tried adding a microwave notch filter at the exit of the signal generator, reducing its phase noise by an additional 8dB, but this did not lead to improved cooling limit of the mechanics.

What is the expected / implied distance in the capacitor gap between the cavity and the membrane? This must be possible to calculate from their data, and should be included.

From simulations, we can estimate the measured microwave frequency to correspond to a capacitor gap of about 450nm, see supplementary section 2.1 and Figure S2.

What is the design frequency of the mechanical damper? How is it constructed? It is easy to include for example a photograph of it in the SI.

In Fig. S10, we show a picture of the mechanical damper. We now also include its spring and mass, giving a damper frequency of 0.5 Hz.

Calibration

=====

In the manuscript, the authors have used a thermal noise peak measurement to calibrate their vacuum coupling rate and the final occupation they achieve.

More information should be provided on how this calibration is performed. In particular, in the range of 0 to 1 K, it is likely that the frequency and internal quality factor of the superconducting cavity has changed, and this must be accounted for in a calibration of the mechanical sideband power since it will change the transduction. To support the data in figure 2A, data showing cavity frequency, cavity internal and external linewidths, mechanical frequency, and mechanical linewidth should be plotted, along with representative fits at a selection of temperatures. Surely the authors have this data: it should be included. Also, datasets showing the observed thermal noise peaks should be presented, and, again, ideally uploaded as an open dataset (there is no reason not to).

More specific details on the exact calibration of g_0 should also be included. Was the calibration performed with resonant cavity driving? What was the amplitude of the cavity drive? With such a high Q-factor, strong resonant driving can easily become self-oscillation if the cavity frequency shifts slightly to the red during measurements, giving an incorrect thermal calibration. Presumably the authors took measures to mitigate this: those measured should be described.

In our work, we have performed two types of calibration, to obtain (a) the bath temperature of the mechanical mode at different cryostat temperatures and (b) the vacuum optomechanical coupling rate g_0 .

Most of section 2.2, including figure 3 describes calibration (a). Here we take into account the appropriate cavity and mechanical parameters at each temperature, as well as a small contribution from dynamical backaction, due to the red-detuned drive. Cavity parameters were obtained from cavity fits, an example of which is now shown in Fig. S3, and their variation with temperature is shown in Fig. S7. Furthermore, the plot in Fig. S9A gives the intrinsic mechanical damping rate as a function of temperature: measured at low microwave power, such that it is free from dynamical backaction. In the final analysis, we furthermore take dynamical backaction into account.

This is now explained in the manuscript, saying

At temperatures below ~ 200 mK, dynamical backaction is small but nonnegligible ($\sim 15\%$). This has been corrected for, together with the temperature-dependent microwave and mechanical damping (see supplementary material). Figure 3A shows the resulting thermalization of the mechanical oscillator to the base plate of the cryostat.

The calibration (b) of the vacuum optomechanical coupling rate is briefly summarized at the end of section 2.2, whereas we have added a section S3.4 to the supplementary information that provides many details of this calibration.

These g_0 calibrations are performed with a pump placed close cavity resonance, giving rise to a small amount of backaction due to a small red-detuning. This backaction is measured by taking two ringdowns: one at the power of the mechanical spectral measurement, and one at the backaction-free low power. The ratio of these two ringdown decay times gives the additional backaction cooling which we take into account when calibrating g_0 .

Furthermore, while a thermal calibration is nice, relying only on the assumption that the sample is at some point in thermal equilibrium with the calibrated sensor, and that the analysis has been done correctly, the system itself is also fundamentally overconstrained: for example if one knows the noise floor of the 4K HEMT amplifier at its input reference plane, and makes an assumption about the attenuation between the sample and the HEMT, then one also has a full calibration of the sample power based on the RT noise floor observed in the spectrum analyser. In such capacitor gap cavities, the microwave resonance frequency itself is also a very sensitive measure of the capacitor gap, and also of g_0 if one performs the appropriate electromagnetic simulations. Combining a knowledge of g_0 and knowledge of the power levels reference to the device also uniquely determines the thermal occupation from a trace at a single temperature.

The question then becomes: are these calibrations consistent with each other? The only real unknown in the second calibration is the attenuation between the sample and the HEMT. If the authors take their thermal calibration as correct, what do they obtain for the attenuation between the sample and the HEMT? Is the number reasonable? This type of cross check is an important part of ensuring that the absolute numbers are trustworthy (which is important as people assign so much importance to a difference between 0.8 and 1.2...)

We agree with the referee on the importance of checking the consistency of calibration. We have now done this using two different methods, which we explain in a new section 3.8 in the supplementary information. The new text reads:

In order to verify that the thermal calibration, on which relies the core result of the manuscript, is correct, we performed two separate consistency checks:

- Given the company specified gain and noise temperature of the HEMT at the operating frequency, we calibrated the spectrum data at low power from figure 4 of the main text, using the background noise as a reference. This translates, given the independent knowledge we have of the overcoupling ratio, the cooperativity and the mechanical linewidth, into a measured mechanical occupation, which is associated with a lower temperature than the extracted 80 mK temperature bath. The ratio between those two temperatures is due to the attenuation between the sample and the HEMT amplifier. We found this expected attenuation to be approximately 6.0 dB. Separately, we summed the company-specified losses of all the components, cables and connectors between the sample and the HEMT, and obtained 6.2 dB, with an estimated systematic error of ± 1.5 dB, given the limited applicability of the specifications to our low-temperature setting combined with the large number of components present and the uncertainties in connector losses for instance.
- Alternatively, as stated in the main text (see the end of the section 2.2), the thermal calibration used provides a measurement of the attenuation between the source and the sample. We found this attenuation to be 66.5 ± 1 dB, and separately, summing again the company specified losses of all the components, we found 65.5 dB of attenuation with an estimated systematic error of ± 4 dB.

Those two consistency checks strengthen our confidence in the thermal calibration presented in Fig 3 of the main text.

This reference plane noise calibration analysis is also important in understanding better the noise squeezing they observe: for example, do they observe saturation of their HEMT amplifier with strong pump tones? How big is the sideband noise power? How does it compare to the expected generator sideband noise power? Without the absolute power numbers and attenuation estimates, it is hard to a priori rule these out.

The specifications of our HEMT amplifier (LNF-LNC4_8C) state a 1dB compression point of -12 dBm for a gain of 39 dB, while we work with a maximum power of 10 dBm and an input attenuation of -66.5 dB and an attenuation between the sample and the HEMT amplifier of -6 dB. As a result, we are operating at least 11 dB below the 1dB compression point. We therefore do not believe that we see a saturation of the HEMT.

Minor comments:

=====

In equation 1, it is quite important to emphasize that the temperature T here is the temperature of the intrinsic `_bath_` of the resonator, not the resonator's mode temperature. In particular, cooling the resonator mode to uK temperatures with sideband cooling will not increase its thermal decoherence rate.

We thank the referee for pointing this out and have emphasized this now as follows: T_{bath} [is] the resonator's bath temperature

*n and n_{cavity} *: Figure 3: Please be clear on what the shown PSDs are. Do they correspond to microwave sideband power spectral densities? Or are they mechanical displacement spectral densities? If they are mechanical displacement PSDs, then express them in $[\text{length unit}]/\sqrt{\text{Hz}}$.

We thank the reviewer for noting this typo. The PSDs shown in figure 4 (which was figure 3 in the previous version of the manuscript) correspond to microwave sideband power spectral densities. The labels have been updated.

“-45 dBm at the source”: For the reader this is an arbitrary number. Given that the authors have a thermal calibration, and also know the cooperativity for example, it should be easy for them to translate all numbers to powers referenced at the input port of the sample. All powers should be referenced to a more relevant reference plane (ie. input of sample).

We agree that the reader needs a more absolute reference for the input power. We therefore indicate the input attenuation, that is inferred from the thermal calibration, in the main text:

By comparison with field-enhanced dynamical backaction effects, this allows us to infer an attenuation of 66.5 ± 1 dB between the signal source and the sample.

This number allows the reader to link the source input power to the sample input power. In order to have a translation into the more universal intracavity photon number, we added the following sentence to the manuscript at the end of section 2.2:

This means that for the highest source power of 10 dBm, the power at the device input is -56.5 dBm and the cavity is populated with $3.3 \cdot 10^7$ microwave photons.

“It has been corrected and is presented in Fig. 2A to show the thermalization of the mechanical oscillator to the base plate of the cryostat.” The users should be more specific on exactly what correction has been applied to the data. Again, the raw data should be plotted, and there is no reason not to include it as open data.

For each data point, two ringdowns are performed, as now presented in figure S6B of the supplementary material. One of them is performed at the power at which the data point is acquired, while the other is performed at a sufficiently low power to rule out any dynamical backaction. By taking the ratio of the slope of those two ringdowns, we extract the broadening due to mechanical backaction, that is used to correct the measured mechanical area.

I appreciate that the authors have uploaded a zipfile of the data plotted, even before publication. It would be nice if there was some more documentation of what that data was (a readme.txt for example). A very nice option would be to include code (jupyter notebooks for example) that demonstrate plotting the data. But this is already a nice step (although I did not myself have time to look at the data...).

A Jupyter notebook, named `Generate_figures.ipynb` is now available to plot the figures of the manuscript and the SI. It is located in the same zip folder than the JSON data files.

Reviewer #2 (Remarks to the Author):

The manuscript presents an experimental report on the realization of ground-state cooling of an ultracoherent electromechanical system. Specifically, a microwave resonator is coupled to a soft-clamped silicon nitride membrane with a super-conducting metal pad. In addition to the ground state cooling, the major achievement here is to reach the strong quantum cooperativity regime and to be

able to maintain a quantum coherence time longer than 100 ms, about three orders of magnitude more than known electromechanical systems. It is argued that the results can be of practical significance for quantum memory applications or ultra-high precision force measurements. More fundamentally, the device with such long coherence times is suggested for testing quantum collapse theories. Overall, the paper is well-written. The diagrams and the figures are clear, and their discussions are intuitively explained.

The paper reports a significant technical advancement for modern quantum technologies and fundamental tests of the foundations of quantum physics. It can be recommended for publication after the authors address the following comments.

1. A short discussion of quantum optomechanical Hamiltonian and how its parameters and field operators are related to the current physical system can illuminate a broader range of readers; otherwise, the paper looks too technical and more accessible to experts in its current form. This discussion and model Hamiltonian should be used to justify the single-mode approximation of the model or to discuss if the setup allows for multimode extension (and if so, would it be possible to cool two or more vibrational modes for more interesting quantum low-lying states?)

We agree with the reviewer on the lack of a broader description of optomechanics in this previous version of the paper. We have therefore added the following text in the introduction of the main manuscript:

This allows coupling it to a microwave resonator to implement the standard optomechanical Hamiltonian

$$H_{\text{int}} = g_0 a^\dagger a (b + b^\dagger), \quad (2)$$

as shown in previous works [1]. Here, $g_0/2\pi$ is the microwave frequency shift due to the zero point fluctuation of the mechanical resonator, \hat{a} (\hat{b}) is the photon (phonon) annihilation operator. Under a strong pump, the system is populated by a mean coherent field around which the Hamiltonian can be linearized to:

$$H_{\text{int}} = g_0 \sqrt{n} (\delta a^\dagger + \delta a) (\delta b + \delta b^\dagger), \quad (3)$$

where n is the mean photon number in the cavity, and the annihilation operators are here small displacements around a mean coherent field. In this case, well-established concepts and methods of optomechanics as described e.g. in [3] apply.

We also added a theory section in the Supplementary material, where the whole model is derived. The single mode approximation is valid here since the mode of interest, as shown in figure 1, is spectrally isolated from any other mechanical mode, with a microwave cavity linewidth smaller than the bandgap size. Furthermore, the coupling mechanism is such that any other mode will exhibit much lower coupling rate. A very small cooling effect of other modes is happening. However, this will not affect the cooling of the mode of interest.

2. The authors should make a more convincing discussion of what is practically and fundamentally more challenging in their ultrahigh coherent system relative to that of Ref. [6] to clarify the novelty and impact of their contribution. At the same time, the authors should emphasize the distinction of

their main idea that allows for the strong coupling and cooling in the ultrahigh coherent electromechanical systems, which was not possible with earlier implementations of such systems.

The main advance over the work of Ref [6] lies in the much higher quality factor and therefore coherence we attain. In particular, Ref [6] achieved a quality factor of $Q=764,000$, while in our work it is $Q=1,5$ billion. Correspondingly, the coherence time in Ref [6] is about $270 \mu\text{s}$, whereas in our work it is above 100 ms. The impact of realizing such long coherence times is manifold, as we discuss in section 3, and includes the possibility to replace bulky microwave resonators as a memory for microwave quantum states.

We more explicitly refer to the platform of Ref [6], and highlight the advance we realize in a revised introduction, which reads:

For state-of-the-art electromechanical systems operated at milliKelvin temperatures, typical Q -factors are $< \sim 10^7$ and coherence times are at most 1 millisecond. This applies to a wide variety of systems, including aluminum vacuum gap capacitors [18, 7, 19, 8, 9, 10], metallized silicon nitride membranes [6, 11, 20, 21] and strings [22], quantum acoustic devices [14, 16], as well as piezoelectrically coupled nanophononic crystals [15, 17]. As a notable exception, $Q \approx 10^8$ has been reported for a metallized silicon nitride membrane in 2015 [23]. Its enhanced performance over similar devices [6, 11, 20, 21] might be linked to its particularly low operation temperature (~ 10 mK) and frequency (~ 100 kHz)—which, among other things, can make operation in the simultaneously overcoupled and resolved-sideband regime challenging.

On the other hand, recent progress in the design of mechanical systems has allowed reaching quality factors in excess of 10^9 at Mega- to Gigahertz frequencies [24, 25, 26, 27, 28, 29]. At milliKelvin temperatures, such ultracoherent mechanical devices can reach $t_{\text{coh}} > 100$ ms, some two orders of magnitude beyond the typical performance of state-of-the-art devices (provided excess dephasing [27] is not an issue). However, so far, the mechanics' coupling to microwave modes has either been extremely weak [24], or absent because of lacking functionalization through e.g. metallization [25, 26, 27, 28, 29]. For this reason, these mechanical systems could not yet be harnessed in electromechanics.

A practical challenge associated with our high-coherence membranes, compared to those of Ref [6], presumably lies in a more difficult fabrication. Due to their larger size (including a phononic shield of several millimeter dimensions) the membrane release and especially the flip-chip assembly with a sub-micron gap are very delicate. For example, the latter poses stringent requirements on wafer flat- and cleanliness.

To make the readers aware of this challenge, we have now added a sentence in the discussion section, where we once more explicitly refer to Ref [6]:

Indeed, $g_0/2\pi=7$ Hz and coupling rates well above 100 kHz have been demonstrated in a similar system [6]. The challenge in transferring this result to our system lies in realizing similarly small capacitive gaps in spite of a significantly larger membrane size, posing stringent requirements on wafer flat- and cleanliness.

3. The method in the experiment uses coherent fields in getting the ground state; can the authors characterize the quantum state in more detail? Only the mean number of vibrational phonons is measured in the experiment; how about the higher-order moments and more complete characterization of the phonon distribution and coherence properties? Perhaps some theoretical estimations and justifications can be given instead of measurements for that purpose?

The electromechanical system we are studying in this work is described by a linear Hamiltonian. Also, the inputs of this system are displaced thermal states which are Gaussian. As a result, the outputs of such a system should be Gaussian, and only the first two moments are required to describe them.

4. Can the authors clarify if their system has any potential for non-local nonlinear terms in its model Hamiltonian to make it more directly relevant to quantum collapse models? This sounds a bit ambitious fundamental application of their results but indeed interesting if possible. A review article may be cited on this topic in addition to the Ref. [36], too.

Yes, reference [36] indeed discusses the implications of the continuous spontaneous localization (CSL) model, which is a nonlinear stochastic extension of the Schrödinger equation. It results in an additional force noise acting to diffuse the mechanical momentum (and position). Reference [36] applies this model explicitly to silicon nitride membranes such as ours.

To clarify this, we have rewritten the final sentence, added a review article and another reference on experiments at the quantum/gravity interface, so that it now reads:

Finally, the membranes' extremely long coherence time could enable electromechanical experiments to test fundamental physics. They may, for example, constrain the parameters of collapse models [39], such as the continuous spontaneous localization model (CSL) [40], which is based on a nonlinear stochastic extension of the Schrödinger equation. Testing the effects of general relativity on massive quantum superpositions with such systems has also been proposed recently [41].

Reviewer #3 (Remarks to the Author):

In the manuscript entitled "Ground State Cooling of an Ultracoherent Electromechanical System," the authors describe a new membrane-based electro-optomechanical system that permits efficient coupling to high quality factor phononic crystal membrane modes using circuit-based cavity optomechanical techniques. Since the circuit that they use to parametrically couple to the defect modes has a linewidth (240 kHz) that is significantly narrower than their mechanical resonance (1.5MHz) they are able to operate this system in the side-band resolved regime. The authors then use this system to demonstrate optomechanical cooling of their long-lived phonon modes, driving the system very near its mechanical ground state ($n < 1$). This manuscript describes a new system that represents a significant advance for the field of electro optomechanics, as it permits quantum coherent access to long-lived phonon modes within membrane type systems. The experimental studies appear to have been conducted with a great deal of thought and care. I recommend publication, provided that the authors address the following comments and make some clarifications within the manuscript discussed below.

Comments:

In section 2.2, the authors indicate that above 200 mK the is insignificant, and the system is assumed to be thermalized with the cryostat, whereas below 200 mK back action becomes significant. It would be helpful to the reader to provide further background regarding the various parameters that are likely changing in the system. For example, is the cooperativity changing as a function of temperature? Does the mechanical quality factor change as a function of temperature? I assume that the thermal conductivity continues to plummet for temperatures below 200 mK. It might be useful to include some discussion of this point in either method section or supplementary information.

We thank the reviewer for making us aware of the need for further clarification. It is indeed correct that system parameters, including mechanical damping, and microwave cavity frequency and decay can be affected by temperature. We now provide explicit plots with data of this variation in supplementary figures S7 and S9. These variations are taken into account in the analysis. To explain this to the reader we now write in section 2.2:

At temperatures below ~ 200 mK, dynamical backaction is small but nonnegligible ($< \sim 15\%$). This has been corrected for, together with the temperature-dependent microwave and mechanical damping (see supplementary material). Figure 3A shows the resulting thermalization of the mechanical oscillator to the base plate of the cryostat.

It is also correct that thermal conductivity plummets at low temperature (see e.g. THERMAL PROPERTIES OF SILICON NITRIDE BEAMS BELOW 1 KELVIN, AIP Conference Proceedings 1219, 75 (2010); [https:// doi.org/10.1063/1.3402336](https://doi.org/10.1063/1.3402336)). However, since this is only of indirect relevance to the physics discussed here (lower heat conductivity makes thermalization less likely), we do not explicitly discuss it.

Also, do the authors have other anecdotal evidence to support the notion that the membrane remains thermalized with the cryostat at temperatures around 200 mK? For example, does the superconducting transition temperature of the membrane suspended superconductor coincide with that of the superconductor that is normally anchored to the cryostat (without a membrane)? This additional information could also be useful to include within a supplementary information.

[Yannick]

We see (Fig 2a) that mechanical noise power is proportional to sensor temperature over a wide temperature range above 200 mK

This is the accepted way to establish thermalisation for the mechanical bath temperature, see refs [4, 5, 6]

As for the material temperature, we now provide in figure S9A in the supplementary material a measurement of mechanical decay rate compared to the temperature of the cryostat. We do not see any discontinuity as function of temperature which indicates thermalization of the material to the fridge.

We do not see any signature of the superconducting transition of the aluminum.

FINISHED

Figure 1 and the description surrounding the electro optomechanical device do not quite provide the reader with enough information. In particular, Figure 1 doesn't provide enough detail to understand the circuit geometry and the coupling method. I only understood the coupling method after reading the description in the main text a couple of times and then stumbling across the inset the bottom of figure 4 in the methods section. I think that it would be prudent to include a conceptual sketch of the circuit diagram indicating how the circuit couples to the defect mode in figure 1 along with the micrographs of the fabricated system. Otherwise, it is unclear how coupling is being performed (which is pretty darn important).

We thank the referee for pointing this out. We have extensively modified Figure 1 and its caption in order to improve clarity especially regarding how the coupling is performed. We have also added a schematic of the circuit, including the inductive coupling to a microwave antenna. On that occasion, we have also split off the panels about the mechanical spectrum and ringdown into a new Figure 2.

Question:

It is interesting to see that tunneling-state two level systems are limiting the quality factor of the resonator at low temperatures. I'm curious to know if the authors observe the same tunneling-state two level system signatures with and without superconducting metal within the defect mode. Any observations along these lines might also be useful to the scientific community.

In earlier attempts we worked with non-metallized membranes, but we do not have systematic data on temperature dependence of Q.

However there is a recent preprint from Eddy Collin that shows good electromechanical coupling to a non-metallised silicon nitride string, however no temperature-dependent Q was measured[arxiv: 2110.00228v1]

There is also one preprint from Cindy Regal that shows T-vs-Q for non-metallized membranes. It appears the scaling could be consistent with a T^α power law (in a certain range)[arxiv: 1611.00878v1]

In Fig S9, we plot the mechanical decay rate as function of temperature for the main mode of interest (subfigure A) which is placed in the first mechanical bandgap, and we also plot this decay rate dependence for a mode in the second bandgap (subfigure B). The power law fits gives a larger exponent for the mode in the second bandgap. Due to its greater spatial localisation, this second mode has a larger relative fraction of Aluminum metalisation. Thus we could draw a relation between amount of Al (and two-level systems in its surface oxide layer) and the value of the power law exponent.

This discussion has been added to the supplementary section 3.6:

We measure the mechanics' intrinsic decay rate as function of thermometer temperature in Fig. S9. We fit the data to a power law of temperature $\propto (T/T_0)^\alpha$, with T_0 and arbitrary reference temperature and the exponent α . For the two analysed mechanical modes, the power law relations with $\alpha = 0.63$ and $\alpha = 0.76$ are consistent with mechanical two-level systems (TLS) coupling to the mode of interest and extracting mechanical energy from it[ZCG+19].

We point out that since the mechanical mode in Fig. S9B at 2.671 MHz is located in the second mechanical bandgap, its spatial extend is less that for the 1.486 MHz mode, located in the first mechanical bandgap. Therefore the Al metallisation makes up a larger fraction of its out-of-plane displacement profile. The larger exponent $\alpha = 0.76$ could thus be attributed to the larger amount of Al in the mechanical mode.

Aside from these areas of improvement described in my comments above, the manuscript does an excellent job of describing key features of their experiments and communicating them to a broad audience.

I'm happy to support publication once the authors make some clarifications and additions described above.

REVIEWERS' COMMENTS

Reviewer #1 (Remarks to the Author):

I appreciate the great effort the authors have put into their answers to my detailed questions. I also appreciate the significant modifications they have made to the manuscript, in particular the now high level of detail in the supplementary information. The issues I raised have, to my satisfaction, all been address, and am therefore pleased to recommend the manuscript for publication.

Reviewer #2 (Remarks to the Author):

The replies by the authors to my comments are convincing and clarify my concerns. The revisions in the manuscript sufficiently improve the paper such that I can unconditionally recommend its publication.

Ozgur E. Mustecaplioglu

Reviewer #3 (Remarks to the Author):

I am satisfied with the revisions that the authors have made. I have no further queries, and I support the publication of their manuscript.

Response to the Reviewer's remarks

Reviewer #1 (Remarks to the Author):

I appreciate the great effort the authors have put into their answers to my detailed questions. I also appreciate the significant modifications they have made to the manuscript, in particular the now high level of detail in the supplementary information. The issues I raised have, to my satisfaction, all been address, and am therefore pleased to recommend the manuscript for publication.

Authors' response to Reviewer #1:

We would like to thank the reviewer for taking the time to assess our work and for their constructive remarks.

Reviewer #2 (Remarks to the Author):

The replies by the authors to my comments are convincing and clarify my concerns. The revisions in the manuscript sufficiently improve the paper such that I can unconditionally recommend its publication.

Ozgur E. Mustecaplioglu

Authors' response to Reviewer #2:

We would like to thank the reviewer for their positive comments.

Reviewer #3 (Remarks to the Author):

I am satisfied with the revisions that the authors have made. I have no further queries, and I support the publication of their manuscript.

Authors' response to Reviewer #3:

We would like to thank the reviewer for their supportive comments.